# Biopolymer Textile Structure of Chitosan with Polyester

**DOI:** 10.3390/polym14153088

**Published:** 2022-07-29

**Authors:** Tea Kaurin, Tanja Pušić, Mirjana Čurlin

**Affiliations:** 1University of Zagreb, Faculty of Textile Technology, Prilaz Baruna Filipovića 28a, 10000 Zagreb, Croatia; tea.kaurin@ttf.unizg.hr; 2University of Zagreb, Faculty of Food Technology and Biotechnology, Pierottijeva 6, 10000 Zagreb, Croatia; mcurlin@pbf.hr

**Keywords:** polyester, alkaline hydrolysis, anionic surfactant, cationic surfactant, chitosan

## Abstract

The research deals with functionalization of a standard polyester fabric with biopolymer chitosan, whose premises are multifunctional and favour ecological effects. Due to the incompatibility of synthetic and natural polymers, the chitosan treatment was preceded by alkaline hydrolysis with sodium hydroxide with the addition of cationic and anionic surfactants as promoters. Compatibility of the chitosan with untreated and alkali-hydrolyzed fabrics was performed by analysis of mechanical and physico-chemical properties. The number of characterisation procedures performed required the use of hierarchical cluster analysis (HCA) to identify homogeneous groups or clusters in which similarities and differences between samples are visible. Almost all applied methods and evaluation parameters have shown that alkaline hydrolysis of polyester fabric has the best potential for functionalization with chitosan. Therefore, the addition of surfactants as promoters during alkaline hydrolysis is not necessary in the pretreatment process phase.

## 1. Introduction

Chitosan as a biopolymer possesses numerous benefits: non-toxicity, biodegradability, antimicrobial activity, wound accelerating effects, etc. [1], so alone or with its derivatives can be used in a wide range of applications [2,3,4,5]. It is widely used for the functionalization of textiles such as wool, cotton, viscose and polyester [6,7]. The poor bonding of chitosan with textile substrates is a challenge that is solved depending on the properties of the polymers. Accordingly, environmentally friendly agents, e.g., citric acid and other low-toxic oxidizing agents, are used to support the interaction, e.g., cross-linking of chitosan with textile materials, e.g., wool and cotton and their blends [8]. Modification of materials with chitosan increases the breaking strength and resistance to deformation, increases the possibility of wetting the material and hydrophilicity [9], enhances dyeing properties [10,11], antimicrobial properties [3,7], reduces static charge [12] and can reduce the amount of particles released from the polyester material during washing [5,13,14]. 

The compatibility of chitosan with polyester can be enhanced by functionalization of one or both polymers by chemical or physical methods [15,16]. Alkaline hydrolysis is a well-known modification process that increases reactivity and hydrophilic character of polyesters [9]. The use of strong alkalis and high processing temperatures leads to irreversible changes in the polyester material [9]. The alkaline action is limited to the surface of polyester, as strongly ionised solution components such as sodium hydroxide cannot diffuse into the mass due to the very non-polar properties of this polymer. Sodium hydroxide can hydrolyse the esters in the polymer, which, in some fragments, depending on the processing conditions, increases the number of carboxyl groups [7,9].

This treatment causes peeling, formation of craters or surface damage, reduces the fibre diameter and leads to a loss of material mass, increases the absorption and dyeing properties and reduces static electric charge [17]. 

The disadvantage of the process of alkaline hydrolysis is a considerable loss of mass and a decrease in breaking force, so that negative impacts should be prevailed by the reaction carried out under optimal conditions [18,19]. Although alkaline hydrolysis has been carried out and studied for many years, its kinetics and mechanism have not yet been fully clarified [20]. Kinetics and mechanism depend on the process parameters, especially alkali concentration, temperature and processing time, as well as addition of promoters. The addition of different cationic surfactants as promoters can reduce the concentration of sodium hydroxide and the weight loss of the treated material [21]. The interaction of surfactant molecules with the substrate can lead to a reduction or increase in the reaction rate, or to a change in the course of the reaction [10,14,15,16,22,23,24,25]. 

In this study, the compatibility between chitosan and polyester fabric is investigated. Accordingly, for better adhesion of chitosan onto polyester fabric, three modification procedures were carried out: alkaline hydrolysis and alkaline hydrolysis with the addition of promoters, cationic and anionic surfactants. After alkaline hydrolysis, polyester fabrics were functionalized with chitosan to obtain a biopolymer textile structure of chitosan with polyester. The characterisation of the untreated polyester fabric, the fabric after alkaline-hydrolyzsis fabric, and the biopolymer textile structure of chitosan with polyester was carried out by the analysis of the structural and physico-chemical properties. Hierarchical cluster analysis (HCA) was carried out with the aim of identifying homeogenic groups in which similarities and differences between fabric samples before and after modification.

## 2. Materials and Methods

The modification was carried out on a standard polyester fabric (PES), delivered by the supplier, Centre for Testmaterials B.V., CFT, Netherlands, the properties of which are specified in Table 1.

### 2.1. Chitosan Treatments

Prior to chitosan treatment, the standard PES fabric was modified by alkaline hydrolysis (AH) in a solution of 2% NaOH (aq) supplied by Ivero, Zagreb, with a bath ratio of 1:5 at a temperature of 98 °C for 30 min in a Polymat, W. Mathis apparatus. The second modification was alkaline hydrolysis with the addition of the cationic surfactant BarquatTM50 from the supplier QuatChem, UK, with the chemical composition of alkylbenzylbenzyl dimethylammonium chloride (AH_K) in a concentration of 3 g/L. The third was alkaline hydrolysis with the addition of 3 g/L of the anionic surfactant, Lavotan SE from the supplier CHT Group, Germany (AH_A). All alkaline modifications were carried out according to the same process parameters, followed by two cycles of rinsing with hot water and cold water, and air drying. 

The treatment of the PES fabric with chitosan, Aldrich^®^ Chitosane, which is 85% deacetylated (LMW), was preceded by the preparation of this solution (0.5%) in deionized water with the pH adjusted at 3.6 with HCl (1 mol/L). With the prepared chitosan solution (CH), the untreated (N) and alkali-hydrolyzed (AH, AH_K and AH_A) PES fabrics were modified by padding in the padder of a Benz stenter at a pressure of 12.5 kg/cm. After impregnation, the fabrics were dried at 90 °C for 40 s and cured at 130 °C for 20 s. The fabrics treated in this way were designated CH-N, CH-AH, CH-AH_K and CH-AH_A.

### 2.2. Methods

In order to assess the effect of modifying PES fabrics by alkaline hydrolysis and functionalization with chitosan, the structural and physico-chemical properties were analysed. All analyses were performed in triplicate. 

The surface charge of the standard polyester fabrics before modification, after modification and after functionalization was analysed using the streaming potential method in the SurPASS electrokinetic analyzer (AntonPaar, Graz, Austria), where the polyester samples were the solid stationary phase, while the electrolyte solution, KCl in concentration 1 mmol/L was the mobile phase. From the values of the streaming potential of the polyester fabric placed in the adjustable gap cell (AGC) and the parameters in the system depending on the pH value of 1 mmol/L of KCl, the zeta potential (ζ) was calculated according to the Helmoltz–Smoluchovsky equation [26,27,28].

The sample of standard polyester fabric (N) was analyzed by the Soxhlet extraction in solvent petroleum ether for 4 h at 40–60 °C in order to assess the preparation’s content. After extraction, the content of the removed preparations from the PES fabric was determined gravimetrically according to expressions (1) and (2):(1)$$S=\left(\frac{m_{p}}{m_{t\;after}}\right)\times 100$$
(2)$$m_{p}=m_{t\;before}-m_{t\;after}$$*m_t_*—fabric mass in g; *m_p_*—mass of preparations in g.

The removed preparations of unknown origin were further analysed by Fourier transform infrared spectroscopy (FTIR) using Spectrum 100S FT IR UATR + TG/IR Interface TL8000 (RedShift) (Perkin Elmer, Waltham, MA, USA).

The tensile properties of all fabrics were analysed by the breaking force and elongation at break according to procedure [29] on the Tensolab tensile tester 3000 (Mesdan S.p.A., Raffa, Italy), distance between clamps 200 mm, at a speed of 100 mm/min and pretension 2 N. Tensile properties were characterized by comparison in tensile force at break. 

The thickness of fabric samples was measured by digital micrometre DM 2000, (Wolf, Mainburg, Germany) of high precision up to 0.001 mm, according to procedure [30]. Thickness was measured at 10 positions and average values were calculated and presented.

The method for determining the pH of the aqueous extract of all samples was carried out according to the procedure [31]. The measurement of the pH of the fabric surface with a contact electrode and the pH of the aqueous extract was conducted using the same multimeter, SevenCompact™ Duo S213, (Mettler Toledo, Zagreb, Croatia).

Moisture transfer were measured on conditioned samples of all fabrics with the Moisture Management Tester MMT 290 (SDL Atlas, Rock Hill, SC, USA) [32], a device for testing the ability to manage moisture (Moisture management tester, MMT) and measuring dynamic moisture transfer in textiles. Testing provides the data on: overall moisture management capability, accumulative one-way transport capability, wetting time for top and bottom surfaces, absorption rate for top and bottom surfaces, max wetted radius for top and bottom surfaces, and spreading speed for top and bottom surfaces [32]. For analysis of polyester samples before and after modification and functionalization, a parameter of wetting time (WT) was selected.

The Drop Shape Analyzer DSA30S, (Krűss GmbH, Hamburg, Germany), was used for the determination of water contact angle (WCA). A fabric sample was placed on a table and recorded by a camera directly connected to a computer monitor. A drop of deionized water at a temperature of 20 °C was applied to the surface of the fabric using a syringe. The DSA30S software started the measurement process automatically and immediately after the liquid was applied, while data was recorded as soon as the drop touched the surface of the textile. 

To detect the presence of chitosan on polyester fabrics, a qualitative staining test for chitosan [6] was carried out using the dye (DyStarGroup) Remazol Red RB 133% (C.I. Reactive Red 2) in concentration of 1% The dyeing procedure of the samples in the Polymat laboratory apparatus, W. Mathis, was carried out at 60 °C for 30 min with a bath ratio of 1:50. In the same apparatus, the washing process with the product Kemopon 50, Kemo (2 g/L) at 90 °C was subsequently carried out for 10 min. The rinsed samples were air-dried and observed by a DinoLite digital microscope (Premier IDCP B.V., Almere, The Netherlands), at 50× magnification. The difference between color strength (K/S value) of unmodified and modified samples was recorded using the spectrophotometer Datacolor 850, (DataColor, Rotkreuz ZG, Switzerland), with constant instrument aperture, standard light D65 and d/8° geometry.

Propensity to pilling was examined by assessing the appearance of samples of reference fabrics and alkali-hydrolyzed polyester fabrics before and after functionalization with chitosan after various cyclic rubs (125; 500; 1000; 2000; 5000 and 7000) on the Martindale Abrasion and Pilling Tester 2561E (Mesdan S.p.A., Raffa, Italy) according to procedure [33]. The sample surface was assessed according to the standard etalons, with a grade of 1 indicating very heavy pilling and a grade of 5 indicating no pilling. 

Surface characterization of all samples pre-coated with a chromium during 120 s was conducted by scanning electron microscopy (SEM) using microscope FE-SEM, Mira II, LMU, (Tescan, Czech Republic) under magnification of 1000× [34]. 

The Fabric Touch Tester, FTT M293 (SDL Atlas, Rock Hill, SC, USA) was used to determine touch properties of polyester fabrics before and after modification/functionalization. Four modules (compression, thermal, bending and surface) are integrated into one piece of equipment and operate at the same time. Primary hand value was characterized by smoothness, softness and warmness, total hand and total touch [35].

The large number of physico-chemical characterization procedures required the use of hierarchical cluster analysis (HCA) [36] with the aim of determining homogeneous groups or clusters whose variables are connected by a certain similarity. HCA was made with the software Minitab, and graphs in the form of Ward’s dendrograms showed the similarities and differences between the observed samples [37]. The results include the formation of clusters, whereby a data set was selected for determining mechanical properties and the analysis of physico-chemical parameters. All groups represent untreated PES fabric, alkali-hydrolyzed fabric with the addition of a cationic surfactant, alkali-hydrolyzed fabric with the addition of an anionic surfactant and all samples treated with chitosan.

## 3. Results and Discussion

Prior to modification by alkaline hydrolysis and functionalization with chitosan, the surface of standard polyester fabric was characterised by zeta potential depending on the pH of 1 mmol/L KCl, as shown in Figure 1. 

Most textile materials possess negative charge in neutral aqueous solutions that is valued by zeta potential. The value of the zeta potential depends on the number and availability of reactive groups [26]. Figure 1 shows the zeta potential curve of the untreated standard polyester fabric. Fewer negative values in the whole pH range are not in line with electrokinetic behavior of standard polyester [38]. The obtained value of zeta potential at pH 10 declined from 69.0 mV as determined for the standard PES fabric of other structural features in the alkaline medium [39]. Therefore, the result obtained shows that preparations were present on the fabric, covering the surface and preventing the complete dissociation—COOH group of the polyester fabric. Therefore, in order to identify the amount of preparations, a gravimetric analysis after 4-h Soxhlet extraction showed that less than 0.1% organic soluble substances were present on the fabric. 

The petroleum ether extract was then analysed by FTIR spectroscopy (Figure 2).

FTIR analysis of the petroleum ether extract confirmed the presence of aliphatic compounds (2851 cm^−1^ and 2918 cm^−1^), a carbonyl group (1738 cm^−1^) and an ethoxylate (1108 cm^−1^). Based on the results obtained, the composition of preparations with a mass content of 0.1% consisted of fatty acid ethoxylates, as well as a small amount of silicone [39,40,41].

Analysis of structural properties is particularly important in assessing the degree of modification, so the characteristics of all polyester samples were monitored by determining the tensile properties and thicknesses. Accordingly, the tensile properties were analysed by measuring the breaking force and the breaking elongation at break of untreated and modified samples of the polyester fabric, as shown in Table 2.

The breaking force of the untreated PES fabric was 1015 N, indicating good integration of the polymer structure. Alkaline hydrolysis of PES fabric reduced the breaking force, and the rate of the decrease depended on the process of performing alkaline hydrolysis. Alkaline hydrolysis with 2% NaOH reduced the breaking force by 5.9%, which was considered an acceptable result. The breaking force of PES fabric in alkaline hydrolysis with cationic surfactant was 556 N, which confirmed the effects on the decrease in tensile properties by more than 50%. Such a high loss is not desirable and indicates the need to optimize the procedure. Alkaline hydrolysis with anionic surfactant did not affect the loss of tensile properties of PES fabric, and the obtained value of breaking force was reduced by 2.1%.

Biopolymer textile structures of chitosan with polyester had different tensile properties compared to all PES fabrics before coating. The standard fabric treated with chitosan had 1.6% higher breaking force than the PES fabric, whereby the chitosan applied to the surface strengthened and improved its tensile properties, which is in line with previous research results [42,43]. Functionalization of alkali-hydrolyzed PES fabrics with chitosan improved tensile properties and increased the average breaking force value by 3.8%. Textile structures of chitosan and alkali-hydrolyzed polyester with surfactants (CH-AH_A and CH-AH_K) were weakened compared to alkali-hydrolyzed with promoters (AH_A and AH_K), whereby the reduction in breaking force was 11.3% with cationic surfactant and 3.3% with anionic surfactant. Based on these indicators, the structural integrity of polyester functionalized with chitosan (CH-N and CH-AH) structures was improved. The values of tensile forces in MPa of untreated and modified polyester fabrics were calculated without taking into account the application of the cross-sectional area of the tested material, as shown in Figure 3. 

Tensile force in MPa depends on fabric properties and modification degree. Histograms in Figure 3 show a big reduction in tensile force of PES fabric modified by alkali hydrolysis with cationic surfactant before and after functionalization with chitosan. 

In order to determine homogeneous groups or form clusters with similar structural properties of all processed samples, HCA was performed. The results using the Ward dendrogram are shown in Figure 4.

Taking into account the structural properties of the dendrogram shown, the influence of alkaline hydrolysis with cationic surfactant (AH_K) before and after modification by chitosan (CH-AH_K), which were combined in a separate cluster compared to alkaline hydrolysis with anionic surfactant (AH_A) and the remaining chitosan treatments (CH-N, CH-AH, CH-AH_A), becomes visible. It is important to emphasize that these are very small changes and they are insignificant compared to the untreated sample (N).

All PES fabric samples before and after modification were analyzed using the pH value of the aqueous extract, which can reveal the migration potential of some substances from PES fabrics (Table 3).

The pH results of the aqueous extract show that the degree of modification and functionalization affects the values listed in Table 3. The untreated PES fabric (N) has a pH of the aqueous extract of 6.97. After alkaline hydrolysis, the pH increases and reaches a value of 9.09, which is to be expected considering the NaOH treatment process. The aqueous extract of the alkali-hydrolyzed sample with cationic surfactant is slightly lower than that of the alkali-hydrolyzed sample and amounts to 8.35. The alkali-hydrolyzed sample with anionic surfactant has a pH of the aqueous extract of 7.79, confirming that this agent neutralizes excess alkali. Biopolymer structures of chitosan with polyester have a pH of the aqueous extract ranging from pH 6.1 to pH 6.31, which is lower than the values before functionalization with chitosan. This is expected given the acidity of the chitosan solution as described in the experimental part.

In addition, the pH value of the surface of all samples was also analysed (Table 3). The surface pH values of all PES fabric samples are lower than 5.0. These low values can be attributed to the hydrophobic properties of the PES fabric, which has low moisture content, and a surface pH is not adequate for the characterization of samples [44].

Several factors contribute to the hydrophilicity of fabrics: (i) increasing the number of hydrophilic groups on the fibre surface; (ii) increasing the surface roughness; (iii) availability of hydrophilic groups on the fibre surface; (iv) increasing the porosity of the hydrolyzed substance. The contribution of these factors depends on the degree of hydrolysis [21]. The differences between the samples before and after modification and functionalization by the MMT process based on the wetting time of the front and back side are shown in Table 4. 

The results in Table 4 show that the wetting time of the untreated PES samples is the highest compared to the modified and functionalized fabrics. There were no major differences found for all treated fabrics. The mean (m) values of water contact angles (WCA) calculated from the left (l) and right (r) angles are presented in Table 5. 

According to the values of the water contact angles, individual samples of analysed fabrics can be grouped according to modification and functionalization levels. Untreated samples before and after chitosan treatment have the highest values (WCA > 40°). This result deviates from the WCA of polyester fabric of other structural features, which is 89.1° [17]. Alkali-hydrolysed polyester fabrics before and after treatment with chitosan belong to the second group have an angle higher than 20°. Untreated and alkali-hydrolysed samples functionalized with chitosan have slightly higher values compared to the initial ones. The third group consists of samples whose contact angle is 0 (AH_K, AH_A, CH-AH_K, CH-A_A).

The degree of modification of the polyester fabric by treatments with alkaline hydrolysis and functionalization with chitosan was recorded by the zeta potential depending on pH 1 mmol/L KCl (Figure 5).

The zeta potential of untreated polyester fabric (N) is previously described. The zeta potential values of alkali-hydrolyzed fabrics (AH), alkaline hydrolyzed fabrics with anionic surfactant (AH_A) and alkaline hydrolysed fabrics with cationic surfactant (AH_K) were shifted to more negative values compared to the untreated fabric. This electrokinetic state is a result of more accessible carboxyl groups in modified fabrics.

The zeta-potential values of the alkali-hydrolyzed sample, and alkali-hydrolyzed sample with anionic surfactant, in the whole pH range are almost identical, while the zeta potential curve of the alkali-hydrolyzed sample with cationic surfactant indicates fewer negative values in the whole pH range, with the IEP of this fabric shifted towards a higher pH. 

The zeta potential curves of the chitosan-modified polyester fabrics in the range of pH less than 8.0 are positive. This surface state confirms the presence of positively charged chitosan on the surface of the untreated and alkali-hydrolyzed polyester fabric in all variants. Based on these relationships, the streaming potential method makes a valuable contribution to the characterisation and identification of the biopolymer textile structure of chitosan with polyester as well as the degree of modification by alkaline hydrolysis.

For a clearer definition of the influence of alkaline hydrolysis and chitosan treatments, HCA was carried out on a set of the obtained results of mechanical properties and the results of physico-chemical analysis. The results in the form of Ward dendrograms showed the similarities of the individual treated samples in relation to the untreated PES fabric, as shown in Figure 6.

The inclusion of the results of the physico-chemical analysis in the HCA shows the individualisation of the samples treated with chitosan as well as the differences becoming visible when PES fabrics are treated by alkaline hydrolysis. The previously presented results of the analyses, indicating a change in the properties of the PES fabric after all treatments, were confirmed by HCA (Figure 5). It can be seen that the samples treated with alkaline hydrolysis belong to the same cluster in all variants, while the samples treated with chitosan are in a separate cluster. Here, a great similarity between the samples is noticeable and the next cluster was formed from both treatments and a significant difference to the untreated material is visible [38]. 

The results show the modification of the polyester fabric by functionalization with chitosan, confirming that the properties of the biopolymer textile structure of chitosan with polyester have changed in comparison to the untreated PES fabric. The results are in line with previous publications [10,11]. 

The identification of the chitosan on the textile structure surface was performed by a staining test with the dye Remazol Red RB. The results of the tests carried out are shown in Figure 7, where the staining of the biopolymer textile structure proves the presence of chitosan. Samples of all PES fabrics after staining were recorded using a digital microscope at 50× magnification and are shown in Figure 7. The color strength of stained samples was expressed by K/S value.

Sulfonated groups of the reactive dye are adsorbed on the positively charged surface of the chitosan-polyester structure by electrostatic attractive interactions, whereby staining proved the presence of chitosan [10]. Red coloration of all chitosan-polyester textile structures (CH-N, CH-AH, CH-AH_K, CA-AH_A) proves the presence of chitosan. The intensity of staining, valued by color strength (K/S), depends on the pretreatment conditions, so the most intense color is obtained in sample CH-AH_K, and the weakest in sample CH-N. Color strength of untreated and alkali-hydrolyzed polyester samples is valued by 0, so are not placed under the micrographs of these samples.

The surface of polyester fabrics before and after modification and functionalization were analyzed using a scanning electron microscope under magnification of 1.000×, as shown in Figure 8.

The micrograph of untreated polyester fabric shows a smooth surface. SEM images of a sample of a standard polyester fabric after alkaline hydrolysis are specific; a slight peeling effect on the surface is observed. Such surface features are in line with previously published results [10]. The smoothness is retained in the sample that was alkaline-hydrolyzed with the addition of a promoter, cationic and anionic surfactants. Promotors have polished most of the hydrolyzed part from the surface. 

The surface of the fibres is uniform and no cracks were observed on the surface. Chitosan treatment is visible on all samples; the fibers are wrapped and coating uniformity is specific for all samples. 

Staining test and SEM micrographs show the presence of substances on the surface. Chitosan red staining of the treated fabrics confirms that this biopolymer is coated on the PES fabric. Intensity and uniformity of staining also show differences between samples, and the best effect was obtained on chitosan-alkaline-hydrolyzed PES fabric, CH-AH.

Due to the importance of tactile properties, untreated, modified and functionalized polyesters were analyzed with FTT, separating the results of the objective evaluation of the surface in terms of smoothness, softness, warmth and total touch (Table 6).

The tabulated results show that the standard PES fabric has the highest smoothness that can be associated with preparations, especially the silicone product, which is evident from the FTIR spectrogram, as shown in Figure 2. It can be seen that the smoothness is impaired through surface treatments by alkaline hydrolysis. The softness of the fabric changes with the degree of treatment and it is evident that alkaline-hydrolyzed samples with the addition of surfactants have better softness compared to other samples. Chitosan treatment of all PES fabrics reduces softness and its grades are equal (grade 1). Grades of total touch of chitosan-functionalized fabrics are also reduced compared to grades before treatment. 

Since the influence of chitosan on fabric touch was confirmed, the fabrics’ propensity to pilling [45] was performed with different abrasion cycles, and the results of the assessment of the surface appearance are listed in Table 7. It is important to look at the obtained results from the aspect of the release of micro-fibrillar (MFs) formations from the surface during the washing process [13].

Table 7 shows the surface grades whereby a favorable influence of chitosan on alkaline-hydrolyzed samples can be seen. The surface grade of the untreated fabric after 7000 cyclic rubs is 3/4, which is confirmed by the FTIR analysis, as mentioned above, due to the composition of the finishing auxiliaries. The alkali-hydrolyzed sample with anionic surfactant before and after modification with chitosan possesses the best propensity to pilling.

## 4. Conclusions

In the presented research, a standard polyester fabric was modified by alkaline hydrolysis processes and functionalized with biopolymer chitosan. To monitor the properties of the biopolymer textile structure of chitosan with polyester, methods were used to analyse the mechanical and physico-chemical properties, breaking force and elongation at break, tensile force, thickness, zeta potential, pH of the aqueous extract, surface pH, moisture transfer, contact angle, morphological characteristics, staining test, touch and pilling of fabrics.

Zeta potential has proven to be the most appropriate parameter for the characterization of the biopolymer textile structure of chitosan-polyester. All biopolymer textile structures of chitosan with polyester have a positive surface charge. The parameters for the characterisation of mechanical properties have confirmed that the biopolymer structure of chitosan-untreated polyester and chitosan-alkaline hydrolyzed polyester shows good tensile properties.

Similarities and dissimilarities among the treated samples and comparison with the untreated PES fabric were determined by hierarchical cluster analysis based on the parameters characterising the mechanical and physicochemical properties of all tested samples. The analysis showed that the samples treated with alkaline hydrolysis belong to the same cluster in all variants, while the chitosan-coated samples belong to a separate cluster.

The identification of chitosan on coated PES textile structures by the Remazol Red RB staining test has proven useful and may be qualitatively and quantitatively preferred over other methods. The touch properties of the biopolymer textile structure of chitosan with polyester were impaired, while the moisture management (MMT) properties were improved in relation to the untreated standard polyester fabric. 

Almost all implemented methods and measurement parameters showed that the alkaline hydrolysis of the PES fabric is a sufficient preparatory stage for functionalization with chitosan, and it is not necessary to add surfactants as promoters in the pre-treatment modification process.

## Figures and Tables

**Figure 1 polymers-14-03088-f001:**
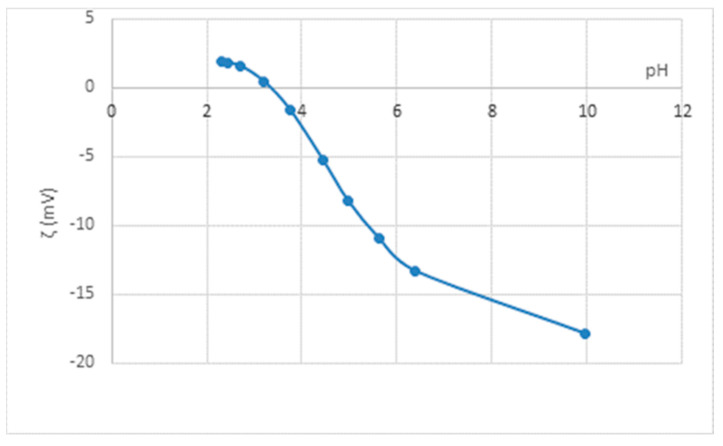
Zeta potential (ζ) of untreated standard polyester (N) fabric in variation of pH of 1 mmol/L KCl.

**Figure 2 polymers-14-03088-f002:**
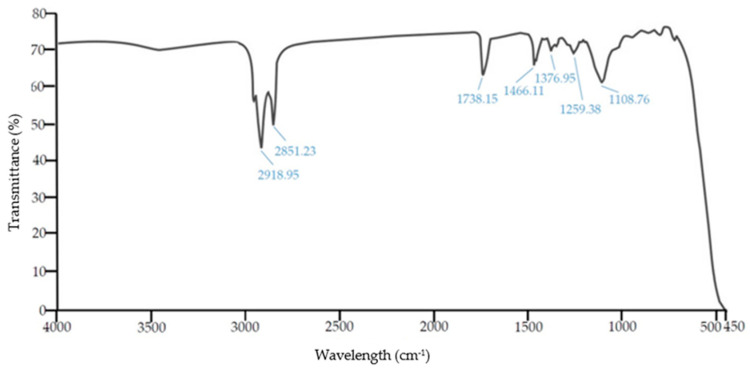
FTIR of petroleum ether extract.

**Figure 3 polymers-14-03088-f003:**
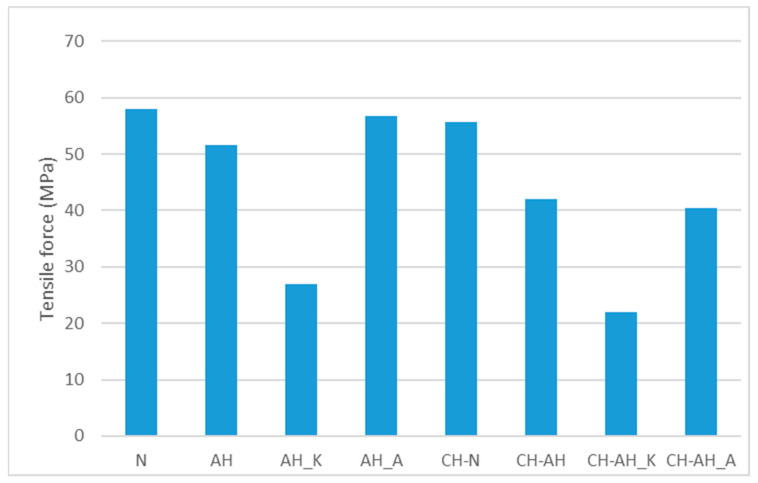
Tensile force in MPa of untreated, modified and functionalized fabrics.

**Figure 4 polymers-14-03088-f004:**
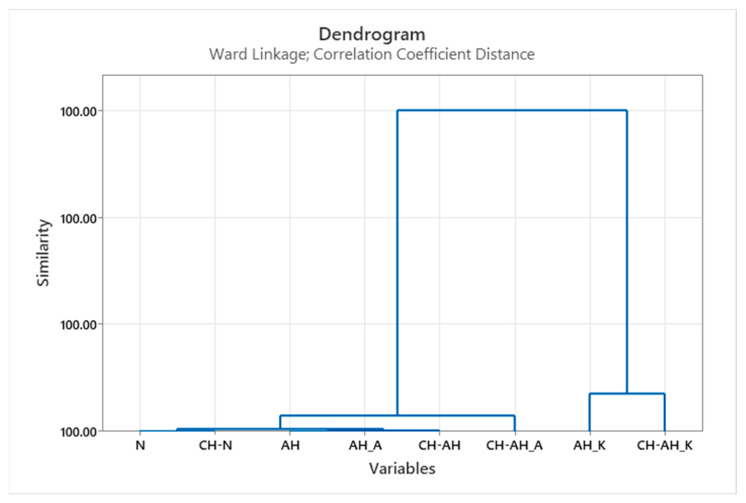
Dendrogram of the HCA according to Ward for similarities/dissimilarities of samples based on data set of structural characteristics.

**Figure 5 polymers-14-03088-f005:**
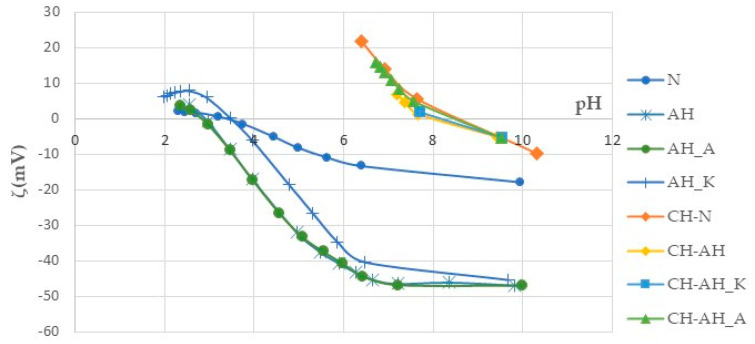
Zeta potential (ζ) of PES fabrics before and after modification by alkali hydrolysis and functionalization with chitosan in variation of pH of 1 mmol/L KCl.

**Figure 6 polymers-14-03088-f006:**
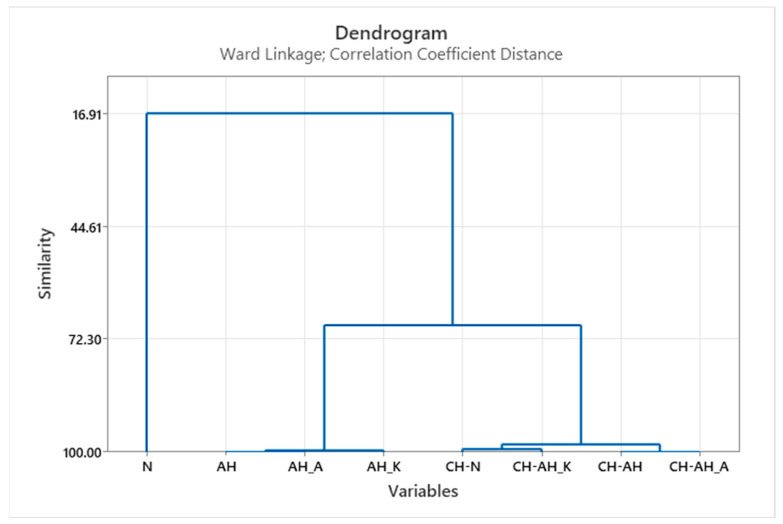
Dendrogram of the HCA according to Ward for similarities of samples based on data set of mechanical and physico-chemical characteristics.

**Figure 7 polymers-14-03088-f007:**
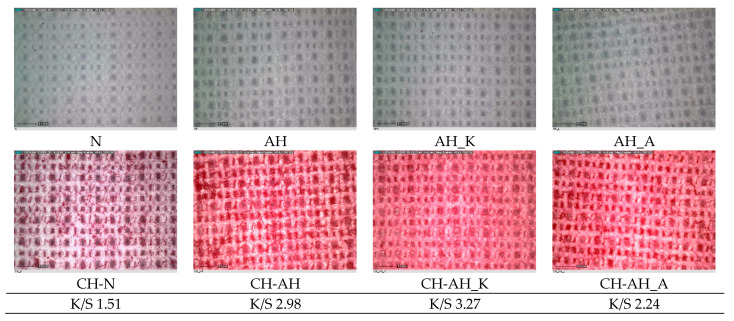
Digital micrographs and K/S value of polyester fabrics before (N) and after modification with alkali (AH, AH_K, AH_A) and chitosan (CH-N, CH-AH, CH-AH_K, CH-AH_A) after staining test under magnification 50×.

**Figure 8 polymers-14-03088-f008:**
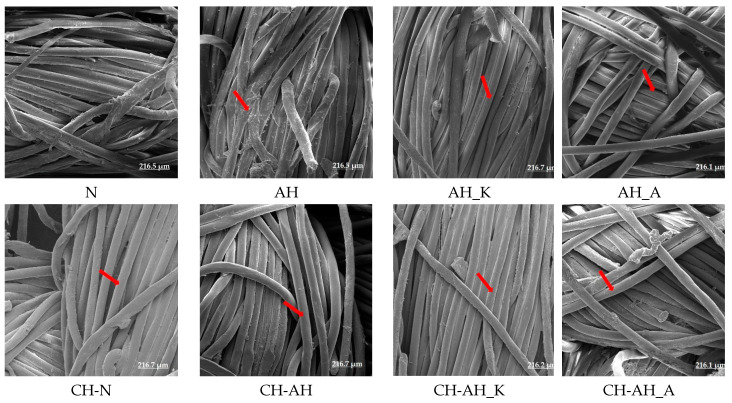
SEM micrographs of polyester fabrics before and after modification with location of changes marked by arrows.

**Table 1 polymers-14-03088-t001:** Characteristics of polyester (PES) reference fabric.

Label	PN-01
Color	White
Mass per unit area (g/m^2^)	156.0
Density (threads/cm)	
ends	27.7
picks	20
Counts of yarn (tex)	
warp	30.4
weft	31.9
Weave structure	Plain weave

**Table 2 polymers-14-03088-t002:** Breaking force (Fp), elongation (ε) and thickness (d) of PES fabrics before and after modification by alkaline hydrolysis and functionalization with chitosan.

Sample		N	AH	AH_K	AH_A	CH-N	CH-AH	CH-AH_K	CH-AH_A
Fp (N)	Mean (N)	1015	955	459	993	1031	987	407	932
	σ (N)	37	50.27	44	61	31	17.85	48	78
	CV (%)	3.393	5.264	9.516	6.145	2.978	1.808	11.765	8.376
ε (%)	Mean^−^ (%)	18.660	18.240	11.370	18.960	19.320	19.530	11.880	19.827
	σ (%)	0.457	0.567	0.707	0.784	0.652	2.149	1.389	1.419
	CV (%)	2.451	3.110	6.216	4.134	3.38	11.002	11.689	7.157
d (mm)	Mean^−^ (mm)	0.35	0.37	0.34	0.36	0.37	0.47	0.37	0.46
	σ (mm)	0.0055	0.0071	0.0055	0.0055	0.0045	0.0436	0.0141	0.0619
	CV (%)	1.55	1.91	1.63	1.50	1.22	9.27	3.82	13.34

**Table 3 polymers-14-03088-t003:** pH of samples aqueous extract (pHae) and surface (pHs).

PES Tkanina	pHae	pHs
N	6.97	4.96
AH	9.09	4.04
AH_K	8.35	4.81
AH_A	7.79	4.86
CH-N	6.03	4.76
CH-AH	6.20	4.83
CH-AH_K	6.31	4.88
CH-AH_A	6.19	4.80

**Table 4 polymers-14-03088-t004:** Wetting time (WT) of face- and back-sided samples.

Sample	N	AH	AH_K	AH_A	CH-N	CH-AH	CH-AH_K	CH-AH_A
WT _face side_ [s]	3.457	2.875	2.519	3.031	1.107	1.557	2.422	1.991
WT _back side_ [s]	3.269	2.687	2.275	2.634	0.975	1.481	2.347	1.822

**Table 5 polymers-14-03088-t005:** Mean water contact angles (WCAm) of the polyester fabrics before and after modification/functionalization.

Sample	N	AH	AH_K	AH_A	CH-N	CH-AH	CH-AH_K	CH-AH_A
WCAm (°)	41.76	21.76	0.00	0.00	43.51	27.78	0.00	0.00

**Table 6 polymers-14-03088-t006:** Smoothness, softness, thermal characteristics and total touch grades of polyester fabrics before and after modification.

Sample	Smoothness	Softness	Thermal Characteristics	Total Touch
N	4.0	2.0	2.0	3.0
AH	3.0	3.0	4.0	3.0
AH_K	3.0	4.0	4.0	3.0
AH_A	3.0	4.0	4.0	4.0
CH-N	2.0	1.0	3.0	1.0
CH-AH	1.0	1.0	4.0	1.0
CH-AH_K	2.0	1.0	4.0	1.0
CH-AH_A	1.0	1.0	3.0	1.0

**Table 7 polymers-14-03088-t007:** Grades of surface after cyclic rubs (125; 500; 1000; 2000; 5000; 7000).

Sample/Rubs	125	500	1000	5000	7000
N	5	5	5	5	3/4
AH	5	5	5	5	3
AH_K	5	5	4/5	4	3/4
AH_A	5	5	5	5	4
CH-N	5	5	5	5	3/4
CH-AH	5	5	5	5	3/4
CH-AH_K	5	5	5	4/5	3/4
CH-AH_A	5	5	5	5	4

## Data Availability

Not applicable.

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
