# Peer review of "Biopolymer Textile Structure of Chitosan with Polyester"

_polymers, 2022, doi:10.3390/polym14153088_

Round 1
Reviewer 1 Report
Manuscript title: Biopolymer textile structure of chitosan with polyester
Manuscript id: polymers-1819330
Authors: Kaurin et al.
The manuscript regarding the topic and results presented is of interest to material/polymer scientific community and revisions based on the comments below are recommended before considering for publication.
Major comments
· Insufficient abstract: In the abstract, the main aim and background of the manuscript is missing, the current version it only highlights the result. In addition, it would be even better to have a sentence as future perspective.
· The unit / abbreviation is not mention before, consider define the abbreviation when mentioned for the first time…. Please check throughout the manuscript to define the abbreviations.
· Line 55-63, the aim or hypothesis of the study is not clear, and the approach is missing ….
· Lake of scientific literature to support the statements and finings throughout the manuscript…... I have made some suggestions for that and more need it….
· More information needed for ALL TABLE captions and define the abbreviation and units that used. And adjust the significant figures for the table and manuscript.
· Grammar and punctuation issuers are need to be addressed. I have selected/mentioned some as example.
· I have a major concern about the results and discussion section. The authors describe results and compare the results with previous studies, however, insight mechanisms are still not sufficient.
· This section is repeating information already presented and explaining things in an unnecessarily complicate way. The quality of the manuscript would benefit from the whole section being condensed.
· The language is generally clear, with some exceptions where the authors are a bit too innovative with the terminology, although there are other good terms to use…..eg ecological profile, that negative influences, streaming potential method, Propensity, . . . .
· In the discussion section, better to discuss the parameters in not-treated material, and the compare it with treated one – now it is opposite.
Minor comments:
Abstract
Line 9: why ‘’multifunctionality’’? this should be ‘’ multifunctional’’
Line 13-16: A complicated sentence, please revise and check the grammar
Introduction:
Line 22-31: You have used 6 references all once. Better to divide the references among the sentences- this is way of referencing is rather confusing. Please correct.
Line 25: ‘’ depending on the properties of the polymers’’ What do you mean exactly? Properties is rather general term.
Line 28-31: Please revise and check the grammar
Line 33-34: Please check the grammar.
With polyester material do you mean ‘’ polyester structure’’
Line 46-47: A reference needed here.
Line 34-41: These are rather long sentence, better to break them down into more sentences.
Line 37-41: A reference needed here. The same for line 47-49, Line 49-50 and 52
Line 49-52: Consider using this reference:
https://doi.org/10.1002/app.10691
or
https://doi.org/10.3390/ma15041530
Line 61-63: Please check the grammar.
In MM section
Literature references are missing for all sub-section. It would be better to cite the references that the procedure adapted.
Section 2.1. Better to separate the list of materials in a separate paragraph, and put this paragraph at the beginning of section 2.1. Put the rest of the text in the following to section 2.1, and named as ‘’ chitosan treatments’’ as you are taking about how you treated the materials
Line 105: what is removed? It is not clear” please clarify …..
Line 119. Move the reference to line 124, so the reference support the treating method not where the material purchased.
Line 125: what do you mean by ‘’ Propensity’’, please clarify to rephrase.
Line 131-133: A reference needed here.
Line 135-7: A reference needed here.
In MM section, what is the quality control (QC) data? There is no mention of the QC.
What is the accuracy of the instruments and recovery. These parameters are needed to report the efficiency of any analytical system.
In general. how many times you’ve recorded the data,? duplicate? Triplicate?..... please elaborate more on this
R&D section
Line 114: Please change ‘’results’’ to ‘’ results and discussion’’ as you have combined them.
Line 147: ……….. titration procedure is not needed it- please rephase to …. By titration of XX with XXX.
Figure 1. What the title for x-axis and y-axis, and more information needed for Figure caption.
Line 150-153: Please check the grammar.
Line 155-158: how exactly this is happening? This is not clear – can you highlight it in the FTIR spectrum?
Line 162-165: Consider using this reference: https://doi.org/10.1002/adv.21629
Table 2: Please explain all the abbreviation? And provide more info for the table title.
Line203-208: this is not clear form the figure 3. Can you highlight it in the figure?
Line 225-231: Please check the grammar. Three opposite conjunctions in one sentence. Please rephrase
Line 232-237: you have started some factors with capital letter while other with small letters – please harmonize throughout the text.
Line 240-244: Please check the grammar.
Figure 2: What the title for x-axis and y-axis, and more information needed for Figure caption.
Line 260-265: A reference needed here. The same for line 280-282, and line 283-285
Figure 3. more information needed for Figure caption, especially the concentration of the staining .
Line 306-311: How exactly ? where can you see this in the figure ? can you highlight this in the figure?
Conclusion
Nice conclusions! However, the future perspectives for following research highly crucial here …..
Author Response
Dear reviewer
please find enclosed answers
Kind regards
Authors

Reviewer 2 Report
Chitosan is nontoxic, biodegradable and biocompatible, and can be used to modify textile materials. The breaking strength, wear resistance, hydrophilicity and antimicrobial properties of the textile material modified by chitosan are improved. Therefore, biopolymer textile structure of chitosan with polyester has very important research significance. In this paper, the textile structure of biopolymer formed by chitosan and polyester is deeply explored around this topic, and the selected topic involves the frontier of disciplines, which has a wide application prospect. With reference to the relevant literature, the author designed the experiment scientifically, with clear logic and sufficient conclusions. However, there are still some issues to be addressed. The specific comments can be found as following:
1. Scientific expression on the affiliations of authors should be applied. Full information, including address should be provided.
2. One sentence should be added at the beginning of abstract to show the background of this research.
3. More introduction on the advantages of chitosan should be further provided with some recent supporting articles: Recent advancements in applications of chitosan-based biomaterials for skin tissue engineering; New Ulva lactuca Algae Based Chitosan Bio-composites for Bioremediation of Cd(II) Ions; etc.
4. More details on the characterization parameters should be provided.
5. The title and content for x axis in Figure 1 should be placed at the bottom of the image.
6. The FTIR spectra should be scientifically modified with better resolution and readability. In addition, the assignment of each peak should be provided with references.
7. The abbreviations in table 2 for breaking force, elongation and thickness should be added into the table title. In addition, scientific expression for data in this table should be applied with average value and standard deviation.
8. The mechanical strength with unit of MPa should be provided and discussed in Table 2 and in the manuscript.
9. How about the wetting behavior of the different samples? Please provide the corresponding contact angle of each sample.
10. The scale bar in Fig. 7 should be remade to have a better readability. Higher magnification images should be provided to clearly show the differences before and after modifications.
11. How to obtain the data in Table 6 should be more clearly clarified in the manuscript.
12. More comparison with previous reports should be performed in the manuscript.
13. The title for 3. Results should be revised into 3. Results and discussion.
14. There are still some typos and grammar issues in the manuscript. Authors should carefully recheck the whole manuscript.
Author Response
Dear Reviewer
Please find enclosed answers
Kind regards
Authors

Round 2
Reviewer 1 Report
I am happy to see the manuscript improved nicely. Author addressed all my comments adequately.
However to make sure the statements are supported with literature I will recommend to add citation in the following lines:
Line 28:
https://doi.org/10.3390/ma14133762
and
https://doi.org/10.1002/9781119441632.ch69
remove one of the full stops.
Line 40-41:
https://doi.org/10.1002/app.41446
Line 45:
https://doi.org/10.1016/j.carbpol.2015.05.077
Overall, the quality of the written text with respect to English phrasing and grammar is good and acceptable.
Best wishes,
Author Response
Dear Reviewer 1
Please find enclosed answers
kind regards
Authors

Reviewer 2 Report
Authors have addressed most of issues well except some minor issues.
1. The reference 1 is about the application of chitosan in textile, which is not suitable for the first sentence to show the bioadvantages of chitosan. It is suggested to replace this reference by the following article: Recent advancements in applications of chitosan-based biomaterials for skin tissue engineering .
2. There is no image for figure 3.
Author Response
Dear Reviewer 2
Please find enclosed answers
Kind regards
Authors
